# Compartmental Cerebrospinal Fluid Events Occurring after Subarachnoid Hemorrhage: An “Heparin Oriented” Systematic Review

**DOI:** 10.3390/ijms24097832

**Published:** 2023-04-25

**Authors:** Fulvio Tartara, Andrea Montalbetti, Emanuela Crobeddu, Daniele Armocida, Eleonora Tavazzi, Andrea Cardia, Marco Cenzato, Davide Boeris, Diego Garbossa, Fabio Cofano

**Affiliations:** 1IRCCS Fondazione Istituto Neurologico Nazionale C. Mondino, 27100 Pavia, Italy; 2A.O.U. Maggiore della Carità University Hospital, Department of Neurosurgery, 28100 Novara, Italy; 3A.U.O. Policlinico Umberto I, Neurosurgery Division, Human Neurosciences Department, Sapienza University, 00185 Rome, Italy; 4IRCCS Neuromed, 86077 Pozzilli, Italy; 5Department of Neurosurgery, Neurocenter of Southern Switzerland, EOC, 6900 Lugano, Switzerland; 6Ospedale Niguarda Ca’ Granda, Department of Neurosurgery, 20162 Milan, Italy; 7Department of Neuroscience Rita Levi Montalcini, Neurosurgery Unit, University of Turin, 10095 Turin, Italy

**Keywords:** subarachnoid hemorrhage, heparin, neuroinflammation, cytokines, delayed cerebral ischemia, vasospasm, blood brain barrier

## Abstract

Subarachnoid hemorrhage (SAH) represents a severe acute event with high morbidity and mortality due to the development of early brain injury (EBI), secondary delayed cerebral ischemia (DCI), and shunt-related hydrocephalus. Secondary events (SSE) such as neuroinflammation, vasospasm, excitotoxicity, blood-brain barrier disruption, oxidative cascade, and neuronal apoptosis are related to DCI. Despite improvement in management strategies and therapeutic protocols, surviving patients frequently present neurological deficits with neurocognitive impairment. The aim of this paper is to offer to clinicians a practical review of the actually documented pathophysiological events following subarachnoid hemorrhage. To reach our goal we performed a literature review analyzing reported studies regarding the mediators involved in the pathophysiological events following SAH occurring in the cerebrospinal fluid (CSF) (hemoglobin degradation products, platelets, complement, cytokines, chemokines, leucocytes, endothelin-1, NO-synthase, osteopontin, matricellular proteins, blood-brain barrier disruption, microglia polarization). The cascade of pathophysiological events secondary to SAH is very complex and involves several interconnected, but also distinct pathways. The identification of single therapeutical targets or specific pharmacological agents may be a limited strategy able to block only selective pathophysiological paths, but not the global evolution of SAH-related events. We report furthermore on the role of heparin in SAH management and discuss the rationale for use of intrathecal heparin as a pleiotropic therapeutical agent. The combination of the anticoagulant effect and the ability to interfere with SSE theoretically make heparin a very interesting molecule for SAH management.

## 1. Introduction

Subarachnoid hemorrhage (SAH) is a severe acute event consisting of massive blood extravasation in arachnoidal spaces and basal cisterns from an aneurysm or arterial branch rupture. SAH accounts for 5 to 10% of strokes with an incidence of 7 to 9 cases per 100.000/year [1]. SAH is more frequent in women and in the age range between 40 and 65 [2] representing 27% of strokes occurring before age 65. Clinical presentation range widely from sudden headaches to drowsiness and neurological impairment until coma. SAH is still burdened by high mortality, ranging from 35 to 50%, and disability rates [1,2]. Poor patient outcome is related to early brain injury (EBI) and subsequent secondary delayed cerebral ischemia (DCI) [3]. Moreover, the development of shunt-related hydrocephalus is an additional cause of damage. EBI occurs in the first period after SAH and is related to increased intracranial pressure and subsequent reduced cerebral blood flow leading to potential global ischemia [4]. DCI is mainly driven by the following secondary events (SSE) triggered by extravasated blood and EBI-induced biochemical changes: neuroinflammation, excitotoxicity, vasospasm, blood-brain barrier (BBB) disruption, oxidative stress, and neuronal apoptosis [5]. In particular, vasospasm, defined as the delayed narrowing of intracranial arteries leading to further ischemic damage, has been extensively studied over the years and different therapies have been addressed to its management [6]. The occurrence of SSE might ultimately lead to irreversible brain damage and poor outcome. Indeed, despite improvement in management strategies and therapeutic protocols, surviving patients frequently present neurological deficits with neurocognitive impairment observed in up to 56% of survivors [2].

A better understanding of the pathogenetic mechanisms mediating SAH at a molecular level is needed to prevent complications such as DCI. Many molecules and mediators that were biologically active within the cerebrospinal fluid (CSF) have been studied in relationship with the occurrence of SSE [7]. Furthermore, several preclinical experimental studies have focused on molecular and cellular mechanisms associated with SAH pathophysiological cascade leading to DCI [8]. The increasing knowledge on this topic has led to the development of natural and synthetic molecules as potential therapeutic treatments [9,10]. However, related to the complexity of SAH-induced events, it remains very difficult to find drugs able to improve patients’ outcomes, and to date only nimodipine has been proven to be effective [11]. Further treatments are needed to effectively prevent neuroinflammation, vasospasm, and events occurring in the subarachnoid spaces. Some studies have reported a positive effect of intravenous unfractionated heparin on the outcome of patients with SAH [12,13,14,15]. Heparin has strong anti-inflammatory effects with many possible mechanisms and various neuroprotective interactions [16]. The aim of this paper is to offer a practical overview of the documented pathophysiological events following SAH occurring in the CSF in humans and related to the development of DCI. The current review highlights the compartmental nature of these events, suggesting the opportunity to test the feasibility of intrathecal compartmental therapy. However, the subarachnoid space is easily accessible via ventricular or lumbar subarachnoid catheters which are commonly used in SAH patients. No information about heparin activity in cisternal compartments is available to our knowledge. We discuss the rationale for the possible use of intrathecal heparin therapy for SAH and hypothesize a way to clinical experimentation.

## 2. Materials and Methods

The Materials and Methods should be described with sufficient details to allow others to replicate and build on the published results. Please note that the publication of your manuscript We performed a review of the literature by analyzing all reported studies regarding the mediators involved in the pathophysiological events following subarachnoid hemorrhage occurring in the CSF with the aim of highlighting the compartmental nature of these events to suggest the opportunity of intrathecal compartmental therapy.

Eligibility criteria: An extensive systematic search has been conducted on Pubmed according to the Preferred Reporting Items for Systematic Reviews and Meta-analysis (PRISMA) guidelines. Our target was to define the mediators and pathophysiological events following subarachnoid hemorrhage occurring in the CSF in humans and related to the development of DCI by analyzing all studies reported in the relevant literature.

Searching for relevant studies, the reference section of included articles was analyzed.

Therefore, while screening the literature, we adopted the following inclusion and exclusion criteria:
-Meta-analysis, Case series, or Clinical study reporting cases of subarachnoid hemorrhage with measurement of CSF inflammation mediators involved in the pathogenesis of DCI.

Conversely, we excluded studies with the measurement of inflammation mediators only in the serum and papers written in languages other than English.

The English literature was systematically investigated using MEDLINE, the NIH Library, Pubmed, and Google Scholar. The last search date was February 2023.

The search was performed by typing the following items:(Subarachnoid hemorrhage, CSF) AND (hemoglobin degradation products), 106 articles;(Subarachnoid hemorrhage, CSF) AND (platelets), 43 articles;(Subarachnoid hemorrhage, CSF) AND (complement), 18 articles;(Subarachnoid hemorrhage, CSF) AND (cytokines), 117 articles;(Subarachnoid hemorrhage, CSF) AND (chemokines), 15 articles;(Subarachnoid hemorrhage, CSF) AND (monocytes), 16 articles;(Subarachnoid hemorrhage, CSF) AND (leucocytes), 55 articles;(Subarachnoid hemorrhage, CSF) AND (endothelin-1), 39 articles;(Subarachnoid hemorrhage, CSF) AND (NO-synthase), 27 articles;(Subarachnoid hemorrhage, CSF) AND (ostepontin), 28 articles;(Subarachnoid hemorrhage) AND (excitotoxicity), 55 articles(Subarachnoid hemorrhage, CSF) AND (matricellular proteins), 36 articles;(Subarachnoid hemorrhage, CSF) AND (blood-brain barrier disruption), 61 articles;(Subarachnoid hemorrhage, CSF) AND (microglia polarization), 17 articles;(Subarachnoid hemorrhage, CSF) AND (heparin), 73 articles.

The first step of selection was focusing on the mediators and pathophysiological events following subarachnoid hemorrhage occurring in the CSF in humans. As a further criterion of inclusion, we chose to consider.

Given these premises, we selected papers according to the following inclusion criteria:-Availability of full-text articles-English text only-Clinical studies with patients older than 18-year-old with a history of SAH

Conversely, the exclusion criteria were:-Full-text articles in languages other than English-Studies not referred to pathophysiological events-Patients younger than 18-year-old

The search returned a total of 703 papers, including molecular and clinical studies. To this initial cohort, the aforementioned exclusion criteria were applied, accordingly eliminating a total of 465 papers that were excluded because they non refer to pathophysiological events occurring after subarachnoid hemorrhage.

The resulting 238 papers are included in our analysis. 55 articles are subsequently excluded after the complete revision of the paper (Figure 1).

## 3. Results

We list the role of several mediators and biologically active molecules that have been studied in relation to pathophysiological events secondary to SAH (SSE) in the CSF compartment. Events occurring after SAH are summarized in Figure 2. The increased pathophysiological understanding of brain damage secondary to SAH opens up new therapeutic potentials. Furthermore, we analyze the role of unfractionated heparin in SAH management.

### 3.1. Hemoglobin Degradation Product and Platelet

Hemoglobin (Hb) and its degradation products have traditionally been considered to be toxins involved in the generation of vasospasm and DCI [18,19]. However, the related physiopathological mechanisms have only recently been clarified [20,21,22]. Cell-free Hb tetramers are released within the CSF from lysed red blood cells (RBC). Dimers deriving from the dissociation of Hb tetramers can then migrate in the CSF, targeting specific brain regions and cerebral arteries. Hb may induce vasoconstriction through scavenging of nitric oxide (NO), generates high lipid oxidant activity, and participates in non-ischemic neuronal damage. Hb toxicity in CSF is related to the downstream heme-Hb metabolite as experimentally demonstrated by malondialdehyde production, the final step of lipid peroxidation, after incubation with reconstituted lipoprotein [23]. Hb might also be involved in the occurrence of microthrombi in distal vessels. Scavenging of endothelial NO induces microvasospasm and disinhibits platelet adhesion and aggregation resulting in mechanical endothelial damage [24,25,26]. A recent study investigated oxyhemoglobin (oxyHb) release from RBC lysis in CSF after SAH. Cumulative oxyHb exposure reaches a peak between day 3 and day 14 and results significantly higher in patients showing DCI [27]. Akeret et al. reported a very strong association between high Hb levels in the CSF and the occurrence of DCI in SAH patients [23]. Furthermore, several authors demonstrated that Hb levels in CSF are related to a poor functional outcome at 3 months. CSF Hb levels were reported to be very low for the first 3 days after SAH, subsequently increasing to finally reach a plateau on days 9–12. Levels of Hb metabolites have also been studied: bilirubin levels increase on day 1 and reach peak levels between day 3 to 5. Biliverdin levels in CSF also increase from day 4 reaching the peak on day 12. Finally, methemoglobin presents a more delayed increase with a peak on day 11 [23]. The increase of Hb and derived metabolites coincides with a high-risk period for vasospasm and DCI suggesting that CSF Hb represents an upstream mediator of SSE. The delayed increase in CSF Hb concentrations is related to the lysis of RBC as indicated by an increase in CSF erythrocyte glycolytic and antioxidant enzymes (e.g., CA1, CA2, CAT, ALDOA). Hemolysis induces marked and rapid macrophage accumulation within CSF as indicated by the increase of soluble cell surface receptors (e.g., CD163, CD14, CSF1R, and TIMP-1). Macrophages accumulate within the damaged brain tissue aiming at erythrophagocytic and Hb-clearing activity, but the massive release of HB might saturate macrophages, explaining the delayed elevated CSF Hb levels. Increased CSF expression of macrophage CD163, a Hb scavenger receptor involved in blood clearance, was positively and independently associated with better outcomes and reduced CSF bilirubin after SAH [28]. These data confirm that the ability to scavenge Hb products in CSF after SAH has a protective role. Hb-scavenger haptoglobin and heme-scavenger hemopexin have been investigated as potential therapeutical strategies [20,22,23].

Oxyhemoglobin and its metabolites, particularly heme, represent the most relevant source of reactive oxygen species (ROS) after SAH occurrence [27]. They react with hydrogen peroxide and produce hydroxyl radicals leading to the subsequent production of lipids ROS from cellular membranes [29]. SAH-induced hypoxia and secondary cell metabolism disruption increased oxygen radicals production and generated a condition of oxidative stress (OS) that refers to an imbalance (disequilibrium) between antioxidant agents and ROS production [30,31,32]. Despite the correlation between OS and outcome is not definitively clarified, OS is the main pathophysiological way leading to intracranial hypertension after SAH [33]. OS-related toxicity induces cell dysfunction through oxidation and alteration of membrane lipids, DNA, and protein eventually leading to programmed cell death [34]. ROS represents a kind of final toxic agent implicated in different SSE and in recent years the evidence that OS plays a crucial role in damage following SAH is growing [35,36]. An intricate antioxidant system physiologically mitigates the free radicals effect. As an example, Superoxide dismutase (SOD) could convert superoxide radicals into hydrogen peroxide consequently reduced by catalase [37]. Despite the cellular antioxidant response being increased after SAH in astrocytes, neurons, and endothelial cells through Nrf2 (Nuclear factor erythroid-derived 2-related factor 2) upregulation, however antioxidant enzymes such as SOD, Glutathione peroxidase (GPX) and catalase are rapidly and heavily consumed reducing significantly antioxidant capacity of brain tissue. This aspect, in addition to the observed depletion of antioxidant molecules such as tocopherols, ascorbic acid, and glutathione, contributes to OS [34]. Decreased SOD concentration in CSF has been correlated with poor long-term outcomes [38].

Recent studies have indicated that microthrombi formation in the distal vessel after aneurysmal SAH (aSAH) is involved in the development of DCI [39]. Such as Hb, platelets extravasated after SAH may play a role in vessel constriction and thrombosis occurrence [40]. Platelets interact with endothelial cells and participate in the inflammation cascade through the release of extracellular vesicles containing chemokines/cytokines [41]. Platelets/microthrombi are found near arteries experiencing vasospasm [42,43]. The mechanism by which platelets induce large artery vasospasm is related to the release of elevated levels of vasoactive thromboxane A2 (TXA2) and Platelet-Derived Growth Factor-b (PDGF) as highlighted by a number of clinical studies [40,44,45,46]. Cisternal level of PDGF-b after SAH has also been reported to positively correlate with the incidence and severity of vasospasm [45]. Platelets have also been considered a therapeutic target for DCI prevention including classical anti-platelet therapies, dual anti-platelet therapies, and platelet activation receptors. Despite promising results, no conclusive evidence is available [40].

### 3.2. Complement in CSF

The complement system represents a major component of induced neuroinflammation after aneurysmal SAH and is composed of membrane-bound regulators and receptors, as well as several plasma proteins. Complement pathways end with the activation of C5 convertase and C5 cleavage, resulting in C5a and the lytic C5b-9 membrane attack complex. C3a and C5a are important proinflammatory mediators implicated in vasoconstriction, activation of coagulation, platelets aggregation, and regulation of tissue factor activity [47]. Van Dick et al. observed that C5a levels in CSF were markedly increased on day 1 after SAH with a gradual decrease within 2 weeks. Furthermore, mice lacking C5a receptors had reduced brain injury, as well as those treated with C5-neutralizing antibodies [48]. These data suggest that C5a is involved in the pathogenesis of brain injury after SAH, despite the lack of association between CSF C5 levels and patients’ functional outcomes. Further and larger studies are needed to elucidate this aspect. However, the involvement of C5 in SSE and its biological role in mediating neuroinflammation after SAH lead to identifying C5 as a potential therapeutical target. Koopman et al. showed with their study on the pharmacodynamic efficacy and safety of Eculizumab in patients with SAH that C5 antibodies may be a promising new treatment option to decrease brain injury occurrence [49]. Moreover, the membrane attack complex formed by C5b-C9 complexes is thought to stimulate hemolysis in the CSF [50]. It also binds cells, such as endothelial cells, ependymal cells, and other brain cells, thereby inducing brain injury. The complement system has been linked to the regulation of synapse numbers [51] and in particular complement components C1q and C3 have been implicated to facilitate the removal of synapses by microglia [52]. Similarly, the initiators of lectin complement pathway (LCP), including ficolin-1, ficolin-2, ficolin-3, and mannose-binding lectin (MBL), have previously been investigated in SAH patients with inconclusive results that overall seem to suggest an association between plasma levels and severity of brain injury [53,54,55,56]. Matzen et al. analysed the time-course of LCP initiator levels in CSF [57]. They found that LCP initiators increased during the first week after SAH and gradually declined over time. Increased CFS levels of all investigated LCP initiators, especially ficolin-1 and mannose-binding lectin (MBL), were associated with a poor functional outcome despite no association being found between delayed cerebral ischemia and overall LCP initiator levels in CSF. Globally, these data suggest that CSF complement activation contributes to brain injury after SAH.

### 3.3. Cytokines in CSF

Cytokines are the most studied inflammatory mediators in CSF after SAH. Many studies have clearly highlighted the role of cytokines in the physiopathology of SAH and DCI. Evidence can be summarized as follows:-An early significant increase of IL-6 might be predictive for the development of symptomatic vasospasm [58] with a strong correlation between elevated CSF IL-6 levels and secondary vasospastic infarcts [59]. Early dynamics of IL-6 in CSF are associated with the outcome of SAH patients [60].-CSF IL-6 seems to be a reliable early marker of vasospasm after SAH on day 3 after treatment before vasospasm clinical onset [61].-CSF IL-6 levels above a cut-off value of 3100 pg/mL are associated with an increased likelihood of ventriculitis; patients with CSF IL-6 levels between 530 and 3100pg/mL are at higher risk for cerebral vasospasm [62].-An increase of intrathecal IL-6 to values ≥ 10,000 pg/mL in the early post-SAH period may be a useful diagnostic tool to predict shunt dependency after SAH [63].-IL-6 receptor antagonist Tocilizumab may be evaluated as a therapeutical agent able to significantly reduce microclot formation, neuronal cell death, and delayed cerebral vasospasm [64].-CSF levels of IL-6 increase over time and are associated with hemorrhage grade. Elevated IL-6 CSF levels may influence SAH progression and may predict poor clinical outcomes in SAH patients. Tumor necrosis factor-α (TNF-α) levels in CSF of SAH patients were higher than those of healthy controls and TNF-α CSF levels increased with disease severity, suggesting that elevated TNF-α levels in CSF may be associated with SAH progression. TNF-α level also correlates with delayed complications of SAH such as DCI [65,66].-Upregulation of H2S-producing enzymes and IL-6 is associated with the inflammatory response and neurological deficits after SAH [67].-TGFβ1 and total TGFβ2 increased significantly in adults following SAH, and there was a significant association between higher CSF total TGFβ1/β2 levels in the acute post-hemorrhagic phase and the subsequent development of chronic communicating hydrocephalus [68].-CSF concentration of Histidine-rich Glycoprotein (HRG) has the possibility to become an early predictor of cerebral vasospasm [69].-Endothelin-1, IL-6, TNF-α, TNFR-I, and IL-1 receptor antagonist (IL-1ra) is elevated in patients with vasospasm [70].-CSF levels of IL-6, TNF-α, IL-17A, IL-10, IL-2, and IFN-γ in the early and delayed phase of aSAH patients were increased as compared to controls. IFN-γ and IL-4 were also increased but did not reach statistical significance. IL-17 is one of the main triggers of the proinflammatory response that could potentially be associated with early brain injury (global ischemia), vasospasm, delayed cerebral ischemia, and increased mortality. IL-17 quantification could be an early prognostic biomarker with clinical value [71]. IL-17 is more closely associated with neutrophil recruitment and activation among the various cytokines. The inhibition of RAR-related orphan receptor gamma T (RoRγt), the master transcription factor of IL-17, decreases the CSF recruitment of neutrophils and could be a therapeutic target to ameliorate DCI [72].-The levels of IL-1β, IL-18, and TNF-α in the CSF were elevated in aSAH patients and were positively associated with cerebral edema and acute hydrocephalus. CSF inflammatory cytokines might be useful biomarkers to assess the severity and predict outcomes [73].

### 3.4. Chemokines in CSF

Chemokines are a family of small cytokines or signaling proteins secreted by cells that induce directional movement of monocytes/macrophages and lymphocytes, as well as other cell types, to sites of inflammation [74]. In their study, Niwa et al. showed an early increase of monocyte chemoattractant protein-1 (MCP-1), a chemokine that stimulates the migration of monocytes, in CSF after SAH [75]. MCP-1 reaches its peak on day 3 after SAH. A high level of MCP-1 has also been found during vasospasm in the major cerebral artery of rats [76] and previous studies have demonstrated the presence of mononuclear leukocytes surrounding the major cerebral arteries in the subarachnoid space after SAH [77]. Interferon-γ-inducible protein-10 (IP-10) and monokine induced by interferon-γ (MIG) were analyzed in the abovementioned study as well. The first one stimulates the migration of monocytes and T cells to inflammatory tissues but does not induce a chemotactic activity in neutrophils. MIG plays its role in activating lymphocytes. Both of them reach their peak on day 5 suggesting a relationship with the inflammatory peak after SAH. Chaudhry et al. focused their study on the chemokine C-C motif ligand 5 (CCL5) showing increased levels on days 1 and 7 [78]. This chemokine is secreted by multiple cells including epithelial cells, endothelial cells, smooth muscle cells, endometrial cells, fibroblasts, platelets, eosinophils, T lymphocytes, glial cells, and neurons [79]. CCL5 is a strong chemoattractant for a large number of inflammatory cells such as basophils, eosinophils, natural killer cells, CD4+ T cells, and CD8+ T cells [80]. Higher CSF CCL5 levels on post-aSAH day 1 are significantly related to chronic hydrocephalus development and DCI. Mohme et al. analyzed 13 pro-inflammatory chemokines in the CSF after SAH, comparing the early (day 1–4) and late (day 6–12) time points. They found significantly increased concentrations of CXCL10, CXCL9, CXCL11, and CXCL1 [81]. This intrathecal chemokine pattern supports the IFNγ-induced immune activation with subsequent chemotaxis of monocytes (CXCL10), T cells (CXCL9, CXCL11), dendritic cells (DCs) (CXCL10), and neutrophils (CXCL1) [81,82]. Increased concentrations of CCL11, CCL2, CCL20, and CXCL1 were associated with the occurrence of delayed cerebral ischemia (DCI). These findings support the role of intrathecal early chemoattractant activation in the pathophysiology of DCI.

### 3.5. Leucocytes and Monocytes in CSF

Translocation of inflammatory leukocytes into an injured area through vascular leakage is a hallmark of the early phase of the inflammatory response. The presence of leukocytes in the subarachnoid space is a specific marker of inflammation [83]. Cells of the innate immune system such as neutrophils, monocytes, and macrophages, have been found in high numbers and in highly activated states in CSF after SAH [84]. Increased CSF concentrations of neutrophils, myeloperoxidase, and NADPH oxidase have been documented in patients with cerebral vasospasm [85]. Myeloxidase (MPO), a heme-containing peroxidase, is released from granules of neutrophils and lysosomes of monocytes recruited in the subarachnoid space and contributes to OS with DNA, protein, and lipids damage related to large production of hypochlorous acid. In SAH patients a correlation between serum levels of MPO and DCI occurrence has been described [86].

On the other hand, little is known about the role of adaptive immunity in aSAH. Roa et al. tried to identify specific immune mediators of cerebral vasospasm after aSAH [87]. They demonstrated that both innate and adaptive immune responses play an important role after aSAH. Innate immune cells enter the subarachnoid space after aSAH. This process occurs via increased expression of cellular adhesion molecules (E-selectin, VCAM-1, ICAM-1, and HMGB-1) and has been correlated with the occurrence of cerebral vasospasm [88,89]. P- and L-selectin start and mediate the leukocyte rolling, the initial event governing leukocyte transmigration from vessel walls into areas of inflammation [90]. Cells of the adaptive immune response have also increased. In particular, CD8+ and CD161+ cells increased significantly in the CSF of patients who developed clinical cerebral vasospasm compared to those who did not. Moraes et al. showed similar results with increased recruitment in CSF and activation of monocytes and neutrophils (innate immune response effectors), but also activation of CD4+ and CD8+ T cells (adaptive immune response effectors) [91]. Coulibaly et al. focused their attention on neutrophil infiltration of the CNS after SAH. Previous data from their laboratory showed that neutrophils in the CSF peak three days after SAH and this peak is predictive of the development of DCI [82]. Mohme et al. analyzed immune cell dynamics in the CSF of 25 patients after SAH. In the CSF, T cells (35–45% of CD45+ cells), granulocytes (35–45%), and monocytes (18%) represented the most prevalent immune cells. Among them, the net cellular influx during the usual onset time of DCI was predominantly accounted for by monocytes and was more pronounced in the DCI group. Leukocyte migration is strongly related to post-SAH inflammatory response and the occurrence of DCI [81].

### 3.6. Endothelin-1 e NO Synthase

Endothelins (ET) are potent vasoconstrictors involved in fluid-electrolyte homeostasis as well as neuronal function [92]. In the central nervous system, ET plays an important role in the regulation of constriction/dilatation of pericytes [93]. There are three isoforms of ET: ET-1 is the strongest endogenous vasoconstrictor across multiple organ systems. Astrocytes are thought to be one of the major sources of ET-1 production within the CNS [94,95]. Several studies have already demonstrated that ET-1 levels are increased in the CSF of patients with SAH [96,97,98]: the study by Cheng et al. showed that ET-1 in the CSF increased in the initial 5 days following SAH, reaching a peak within 3 to 5 days, and then gradually subsided [99]. ET-1-induced vasoconstriction is mediated by two G-protein-coupled receptors: endothelin receptor type A (ETA) and endothelin receptor type B (ETB) [100]. The expression levels of ETA and ETB mRNA are upregulated in CSF following SAH probably leading to hypercontractility of cerebral arteries [98]. Suzuki et al. revealed that ET-1 expression levels in both plasma and CSF of patients with SAH classified as Fisher grade III to IV were significantly higher, compared with those in patients with SAH classified as Fisher grade I or II [99]. The study by Cheng et al. only included patients with SAH Fisher grade III or IV, who were more prone to develop cerebral vasospasm [99]. The peak expression of ET-1 in CSF appeared within 3–5 days and remained at a high level until 10 days after SAH onset. Related to the high risk of cerebral vasospasm development within 4 to 10 days following SAH, it was speculated that ET-1 expression in the CSF may be a potential biomarker to predict cerebral vasospasm following SAH. Wanebo et al. demonstrated that systemic administration of the ETA receptor antagonist significantly attenuates cerebral vasospasm after SAH, thus providing additional support for the role of ET-1 in vasospasm [101].

Nitric oxide (NO) is frequently termed a “double-edged sword” in cerebral ischemia. It is a powerful dilator of cerebral vessels and it has been reported to have both neuroprotective and cytotoxic effects [102]. NO is produced by endothelial nitric oxide synthase (eNOS) in the intima and by neuronal nitric oxide synthase (nNOS) in the adventitia of cerebral vessels [99]. Perivascular OxyHb induces the inactivation of Ca^2+^ channels, and the consequent drop in intracellular Ca^2+^ in endothelial cells leads to reduced eNOS expression [103]. Dysfunction of eNOS could also be due to increased activity of phosphodiesterase (PDE) and endogenous inhibition by asymmetric dimethylarginine (ADMA) [104]. Ng et al. showed high CSF NO levels in 16 patients with spontaneous SAH; the degree of elevation was higher in patients with poor-grade SAH [103]. Woszczyk et al. investigated 21 patients after aSAH and 5 of them developed clinically symptomatic vasospasm. There was a significant difference in NO levels between the groups. Patients with cerebral vasospasm showed between days 2 and 8 significantly higher levels of NO metabolites (nitrate and nitrite) in CSF than patients with an uncomplicated clinical course. Within OS induced by SAH, the reaction of superoxide radicals with NO leads to the production of producing neurons [31]. Consequently, overproduction of NO may cause free radical injury of cells of the vascular wall and induce vasoconstriction [105]. In their recent study, Kho et al. also revealed an association between elevated NO levels in CSF and the severity and occurrence of vasospasm other than clinical outcomes in aneurysmal SAH patients. They suggest that increased levels of NO metabolites detected in vasospasm patients are the result of inducible NO increase secondary to immunological response following SAH. Subsequent overproduction of NO may be the cause of free radical injury of cells of the vascular wall leading to vasoconstriction [106].

Pluta et al. on the other hand described a close relationship between the decrease of CSF nitrite levels and the development other than the severity of vasospasm. Moreover, nitrite infusion prevents vasospasm in the experimental SAH monkey model [107]. They suggested the hypothesis that decreased NO availability is responsible for the development of cerebral vasospasm [108].

### 3.7. Excitotoxicity

The recent literature suggests excitotoxicity, a type of neurotoxicity mediated by glutamate, plays an important role in EBI. Glutamate is the principal neurotransmitter in the adult brain and its role is critical for neuron-to-neuron communication, neuronal growth, and synaptic plasticity in health conditions and diseases [109]. Excitotoxicity consists of two components: the first one is an acute, intracellular influx of Na+ and Cl-, and subsequently water, causing cell swelling and tissue edema that compress the microvasculature in the surrounding regions resulting in microcirculatory disturbances, and the second one is excess in Ca2+ cellular influx Ca2+-dependent activation of death signaling cascades leading to cell degeneration that occurs somewhat late [110,111]. The onset of SAH and post-SAH global cerebral ischemia induces metabolic failure with the disturbance of ionic hemostasis, causing excessive and uncontrolled releases of glutamates and glutamate receptors overstimulation [112]. Increased CSF levels of excitatory amino acids (EAAs) such as glutamate and aspartate have been reported in SAH such as in hemorrhagic traumatic brain injury. [113,114]. Higher levels of EAAs is related to poor outcome contributing to vasospasm and neuroelectric disturbances defined as cortical spreading depolarization (CSD). CSD is a massive focal neuronal depolarization leading to further cerebral ischemia, decrease in NO and K+ increase in subarachnoid space with consequent worsening of brain edema as well as predisposition to seizure activity [115].

Further, post-SAH hemolysis increases basal perivascular K+ concentrations and arginase-1 release that decreases NO availability through depletion of the eNOS substrate L-arginine. NO dysregulation directly led to parenchymal arteriolar constriction rather than arteriolar dilatation irrespective of increased metabolism [116]. Therefore, a correlation between excitotoxicity and the prognosis of the patient with SAH has been suggested. Cerebral glutamate and EAA levels in an acute phase of SAH are already high in patients with neurologically poor status and cerebral edema; elevated glutamate levels with depletion of the eNOS substrates were reported to be an independent predictor of poor outcomes in a clinical setting [112,116]. NMDAR (N-Methyl-D-Aspartate Receptors) present a high affinity for glutamates and are involved in receptor-mediated Ca2+ influx. Inhibitors of NMDAR, such as Ketamine, have been considered neuroprotective agents preventing CSD but conclusive evidence is not still available [117].

### 3.8. Ostopontin and Matricellular Protein

Osteopontin (OPN) is a pleiotropic acidic matricellular glycoprotein involved in the pathogenesis of acute and chronic inflammation [118]. It undergoes upregulation under pathological conditions and can be secreted by microglia under stress [119]. OPN may induce cell motility and modulate the function of bioactive substances through interaction with a variety of mediators such as growth factors, chemokines, and proteases. Production of OPN is stimulated by transforming growth factor-β, PDGF, cytokines such as interleukin-1α or tumor necrosis factor-α and endothelin [120]. OPN has been indicated to be neuroprotective in aSAH [121,122] preventing EBI and vasospasm. Elevated plasma OPN levels have been reported as an independent predictor of poor prognosis at 90 days in patients with SAH. However, to date, only one study reports osteopontin CSF values, confirming that high values with an increase between day 4 and day 8 are associated with a poor prognosis. CSF levels of OPN are higher than plasma levels at any time suggesting a primary production and release in the CSF compartment. Moreover, persistent elevated plasma OPN levels on days 11–12 are associated with chronic shunt-dependent hydrocephalus. Osteopontin has been proposed as a potential therapeutic agent but there are no clinical trials demonstrating its efficacy. The presence of high CSF levels of osteopontin, however, did not show protective effects but was associated with poor prognosis and chronic hydrocephalus. High OPN CSF level probably represents the response to greater damage more than being a molecule able to prevent neurological damage.

### 3.9. Blood-Brain Barrier Disruption

The blood-brain barrier (BBB) is a highly selective semipermeable membrane composed of various interacting cells such as endothelial cells, astrocytes, microglia, and pericytes. Endothelial cells, with their tight junction (TJ) and membrane protein acting as regulating transporter systems, are responsible for the correct barrier functioning [123]. SAH induces alterations in every single component of the BBB cells leading to brain homeostasis disruption. The main pathophysiological event occurring after SAH and related to progressive BBB dysfunction is the development of a neuroinflammatory response. BBB damage can be already observed after 24 h from SAH due to the degradation products of erythrocytes such as oxyhemoglobin (OxyHb) enhancing OS that start the endothelial cell apoptosis [124,125]. Apoptosis in endothelial cells is orchestrated by endoplasmic reticulum (ER) stress-induced activation of C/EBP homologous protein (CHOP). SAH induces increased CHOP levels, which leads to the downregulation of the anti-apoptotic Bcl-2 protein and induction of the Bcl-2 interacting mediator of cell death (Bim) [126]. Moreover, increased levels of key pro-apoptotic proteins such as p53 upregulated modulator of apoptosis (PUMA) and Bcl-2-associated X protein (Bax) were found in endothelial cells 24 h after SAH. Therefore, p53 seems to be one of the key factors in the control of endothelial cell apoptosis following SAH. TNF-α also plays an important role in the apoptosis of endothelial cells after SAH through the action of the TNF-α-receptor that activates caspase-2, -3, -8, and -9. Caspase-8 in turn activates caspase-3, which subsequently cleaves poly (ADP)-ribose polymerase (PARP), resulting in DNA fragmentation and cell death [127].

OS-related production of free radicals can negatively affect several cellular structures, such as membranes, lipids, tight-junction proteins, lipoproteins, and deoxyribonucleic acid (DNA) [128,129]. Lipid membrane peroxidation spread very quickly affecting a large number of cells [125]. Proteins may as well be damaged by oxidative stress, undergoing conformational modifications that could determine a loss, or impairment, of their enzymatic activity [129,130]. DNA modification can induce endothelial cell apoptosis and altogether, these mechanisms increase BBB permeability [131]. Reactive oxygen species are also released by infiltrating neutrophils along with the release of proteases such as elastases, collagenase, and matrix metalloproteinase (MMP) [132].

Another important mechanism that increases BBB permeability is the disruption of the microvascular basal lamina mediated by loss of collagen IV after SAH, due to increased enzymes such as collagenase and MMP-9 [133]. Damages to endothelial cells and the basal lamina are responsible for the development of cerebral vasospasm. These morphological changes have been reported to reach a peak on days 5 and 7 after bleeding which corresponds to the most dangerous period for vasospasm development [134].

Inflammation is an important factor in the progression of BBB disruption. Activation of the NF-κB inflammatory pathway, as well as the increased expression of Toll-like receptor (TLR)-4 and p53, induce the up-regulation of MMP-9 via NF-κB and was recorded in brain endothelial cells only 24 h after SAH. This again leads to the degradation of occludin and disruption of basal lamina through the degradation of collagen IV and laminin [135]. The disruption of tight junctions between endothelial cells is considered to be the main cause of post-hemorrhagic vasogenic edema [136].

### 3.10. Microglia M1 and M2 Polarization

Microglia, the residential immune cells of the central nervous system (CNS), have several functions: antigen presentation, phagocytosis, and expression of cytokines and chemokines [137]. Microglial cells are sensitive to numerous inflammatory mediators, transcription factors, or growth factors and activate promptly by altering the morphology and polarization in response to various brain insults. Conventionally, microglia gain morphology changes from ramified to an amoeboid shape when activated. Activated microglial cells polarize to M1 and M2 phenotypes, which are distinct activation states with different expression profiles [138]. Classically, lipopolysaccharide (LPS) and pro-inflammatory cytokine interferon-γ (IFN-γ) prime microglia to M1 phenotype with the release of pro-inflammatory cytokines, such as TNF-α and interleukin-6 (IL-6); when exposed to IL-4 or IL-13, microglia polarizes to alternative M2 phenotype with the expression of anti-inflammatory factors, such as transforming growth factor beta (TGF-β) and IL-10 [139,140]. Zheng et al. demonstrated that microglia were activated 24 h and 72 h after SAH [141]. Microglia activated dynamically after SAH, starting with an early M1 phenotype and transitioning to M2 phenotype. M1 polarized microglia drives the pro-inflammatory responses against microorganisms and tissue injuries, producing a high level of reactive oxygen species, NOs, and pro-inflammatory cytokines [142]. M2 phenotype is a beneficial activation state characterized by scavenging debris, promoting angiogenesis, and expressing anti-inflammatory factors [143]. A better understanding of these mechanisms would provide a key foundation for further investigations to develop target treatment.

Tian et al. revealed that activation of retinoic acid receptors (RARα) receptor improved neurological outcomes and attenuated neuroinflammation of EBI after SAH by promoting M1-to-M2 phenotypic polarization of microglia (Mafb/Msr1/PI3K-Akt/NF-kB pathway regulation) [144]. The signal transducer and activator of transcription 3 (STAT3) are closely related to the microglial polarization transition and modulation of microglia-dependent neuroinflammation. Microglial STAT3 deletion improved neurological function and neuronal survival probably by promoting M2 polarization and anti-inflammatory responses after SAH. STAT3 might be a promising therapeutic target to reduce EBI after SAH [145]. Recombinant human erythropoietin (rhEPO) acts on EPOR/JAK2/STAT3 signaling pathway and also showed anti-inflammatory effects on microglia polarization reducing brain cell apoptosis, neuronal necrosis, albumin exudation, and brain edema acting on [146]. Gao et al. focused on milk fat globule-epidermal growth factor-8 (MFG-E8), a secreted multifunctional glycoprotein composed of epidermal growth factor (EGF)-like sequences [147]. Given its role as a bridging molecule between apoptotic cells and macrophages, MFG-E8 facilitates the clearance of pro-inflammatory mediators. Recombinant human MFG-E8 treatment mediates the M2 microglial shift and reduces microglial inflammatory response producing direct protective effects on neurons (integrin β3/SOCS3/STAT3 signaling pathway). Therefore, also MFG-E8 is a promising candidate to suppress microglia-mediated neuroinflammation and improve SAH patients’ outcomes [148]. It is very likely that microglial activation plays an important role in brain damage after SAH. Limiting microglial activation with an early reduction of the inflammatory response may be a key therapeutic target.

### 3.11. Neuronal Apoptosis

Biological responses to SAH can lead to the apoptotic death of neurons, glial and endothelial cells. Apoptosis refers to the death of cells under the autonomous control of multiple genes and can occur through three different pathways, namely, the extrinsic pathway, intrinsic (mitochondrial) pathway, and endoplasmic reticulum (ER) stress-induced pathway, depending on the site of apoptosis [149]. In addition, other molecular mechanisms such as oxidative stress pathways can play a main role in apoptosis occurrence. OS induced during the ischemic post-SAH phase contributes to enhancing excess mitochondrial ROS production and mitochondrial dysfunction [37]. It has been suggested that neuronal apoptosis is the major contributor to morbidity and mortality after SAH. Neuronal cell loss probably continues into the later phase and the larger part of cells dying after SAH are neurons [149]. The persistence of blood in the subarachnoid space perpetrates damage to brain cells. Apoptosis can therefore be considered a final effect when irreversible triggered and all the mechanisms previously described may lead to an inflammatory state and microglial activation thereby increasing neuronal apoptosis. Reduction and modulation of SSE represent the main therapeutical target in order to minimize apoptosis.

### 3.12. Heparin and SAH

Unfractionated heparin (UHF) was proposed as multitargeted therapy for the prevention of delayed damage in SAH patients. [150] Only a few studies investigated the use of heparin in SAH management. A retrospective cohort study on 87 consecutive patients with Fisher grade 3 aSAH documented the potential benefit and safety of low-dose intravenous heparin (LDIVH). Patients treated with heparin had a statistically significant reduction in symptomatic vasospasm (9% vs 47%) and CT-documented brain infarction (0% vs 21%) as compared to the control group. [12] Furthermore, the retrospective study by James et al. focused on the cognitive outcome of SAH patients based on the Montreal Cognitive Assessment [151]. Patients treated with LDIVH showed better cognitive outcomes as compared with the control group treated with standard therapy. No severe cognitive impairment occurred in LDIVH patients and the Authors underlined the positive influence of heparin through linear regression analysis. In order to evaluate the neuroprotective effect of UFH, Burder et al. conducted a retrospective study on 718 patients treated for aSAH [13]. The rate of cerebral vasospasm was significantly reduced in patients treated with a continuous infusion of UFH following aneurysm endovascular coiling compared with a control group (14.2% vs.25.4%; *p* = 0.05). Despite no statistical significance, the heparin effect enhanced when the treatment was continued for 7 days. In a more recent retrospective study on 556 patients, LDIVH resulted beneficial as compared with prophylactic subcutaneous heparin. The Heparin cohort was 1.9 times less likely to develop delayed neurological damage and 2.5 times less likely to develop cerebral infarction as demonstrated by multivariate analysis. Finally, the meta-analysis of Lukito et al. confirmed that heparin treatment for at least 48 h is associated with reduced occurrence of brain infarction showing at the same time adequate treatment safety [152]. Wurm et al. analyzed 120 consecutive patients with aSAH (Hunt-Hess I–III), after aneurysm repair, randomly allocated to either one subcutaneous injection of 20 mg enoxaparin or placebo for 21 days following SAH. The study revealed a marked reduction in vasospasm-related infarction (3.5% vs. 28.3%; *p* < 0.001), shunt-dependent hydrocephalus (1.8% vs. 16.7% placebo; *p* = 0.019), and delayed ischemic deficits (DID) (8.8% vs. 66.7% placebo; *p* < 0.001). At 1-year follow-up, patients in the enoxaparin group had significantly better outcomes than the placebo group, supporting the neuroprotective properties also of low-molecular-weight heparin [153].

UFH is a mixture of endogenous glycosaminoglycans with molecular weights ranging from 3 to 30 KD consisting of variable-length linear polymeric chains of heavily sulfated polysaccharides. The high degree of sulfation results in high negative charge density making UFH the highest negatively charged biological molecule existing [154,155]. Within common understanding, the primary effect of heparin is to bind antithrombin III inducing an allosteric activation that allows antithrombin III to inhibit the clotting Factor Xa. The heparin-antithrombin III complex may also bind and inactivate thrombin [156]. Heparin could interfere with the evolution of extravasated clots in the subarachnoid space as well as with platelet aggregation and the release of mediators. Despite the prominence of anti-coagulation in heparin clinical applications, much evidence suggests that anti-coagulation is not the primary physiologic role of heparin. Related to high negative charges, UFH has a strong ability to interact and bind with positively charged proteins and surfaces. UHF has been documented to interact with over 100 proteins [157]. UHF can directly bind and inactivate cytokines, chemokines, and growth factors, and release proteins such as elastase, collagenase, matrix metalloproteinase, and cell-surface glycosaminoglycans. Therefore, heparin has direct anti-inflammatory properties [158,159] linked to the binding of proinflammatory factors [160,161]. Heparin has been shown to specifically decrease the number of leukocytes participating in the inflammatory response to any insult occurring within the CNS [162,163]. In fact, UFH is a potent inhibitor of the adhesion molecules such as both leukocytes and endothelial cell selectins [164]. Consequently, heparin reduces leukocyte adhesion, rolling, and extravasation in CSF in different neuroinflammatory conditions [165,166,167]. Prevention of leukocyte migration is part of the strong anti-inflammatory effect of Heparin. Given the uniform distribution of the negative charges and large size, UFH is able to stoichiometrically bind four oxyhemoglobin molecules neutralizing the toxic effects of free hemoglobin and related free radicals production in the CSF [168]. Furthermore, UFH has a number of anti-oxidant effects demonstrating a strong direct scavenging antioxidant effect neutralizing reactive oxygen species as occurred in several in vitro studies [56,57,169,170,171]. In addition, UHF is able to increase the synthesis and release of extracellular SOD into circulating fluids [172,173]. Heparin administration in rabbit CSF results in a 27-fold increase in SOD activity within the CSF [174]. Heparin is also able to modulate endothelin-1 (ET-1) activity decreasing the transcription of endothelin-1 and ET-1 promoter [41,175] modulated by heparin-binding epidermal growth factor [176,177]. Significant vasoconstriction in vascular smooth muscle cells due to ET-1 is mediated through the epidermal growth factor receptor (EGFR). Heparin-binding epidermal growth factor, a ligand of EGFR, modulates its transactivation [178]. These effects can play a positive role in the prevention of vasospasm.

Heparin may lead to the inhibition of intracellular nuclear factor-kappa B (NF-kB). NF-κB is a transcriptional factor required for the gene expression of many inflammatory mediators, such as IL-1β, TNF-α, IL-6, intercellular adhesion molecule-1 (ICAM-1), and monocyte chemoattractant protein-1 (MCP-1) [179]. Hochart et al. showed that treatment with pharmacological doses of LMWH and UFH significantly attenuated lipopolysaccharide-induced production of TNF-α, IL-8, IL-6, and IL-1b as well as NF-kB translocation. The inhibition of NF-kB activation certainly represents one of the mechanisms by which heparin exerts its anti-inflammatory effect [180].

Finally, studies have also found that heparin can reduce BBB dysfunction and the resulting cerebral edema in traumatic brain injury, ischemic stroke, intracerebral hemorrhage, and meningitis. However, its effect on cerebral edema in patients with aSAH has not been studied [181]. As a negatively charged glycosaminoglycan, Heparin has also been shown to be involved in myelin preservation and inhibition of apoptosis [157,166]. Demonstrated effects of UFH are summarized in Figure 3.

In conclusion, UFH can potentially antagonize most of the pathophysiological pathways occurring after aSAH and thereby could act as a multi-targeted therapy to reduce SSE and DCI [16]. However, the available studies on the use of heparin in ESA are few, and many aspects such as the optimal timing of administration and the real risk of bleeding need to be clarified.

## 4. Discussion

A greater improvement in the functional outcome for SAH patients will probably stem from therapies targeting some of the molecules described so far, in particular, related to a reduction of DCI. To date, however, the trials completed on new drugs in SAH did not show promising results and the only significant evidence concerns nimodipine.

The cascade of pathophysiological events secondary to SAH is very complex and involves several interconnected, but also distinct pathways. The identification of single therapeutic targets or specific pharmacological agents, such as, for example, ET-1 receptor antagonist Clazosentan [9], may be a limited strategy able to block only selective pathophysiological pathways but not the global evolution of SAH-related events, explaining the unsatisfactory results of previous clinical trials. On the other hand, the simultaneous use of several drugs acting on different targets can be difficult due to the accumulation of secondary and side effects. Therefore, the possibility to address research focused on molecules with different mechanisms of action and with the ability to block various pathophysiological pathways may result very interesting [149]. The pleiotropic effect of UFH was highlighted in this review in relation to the described compartmental pathophysiological events. The combination of the anticoagulant effect and the ability to interfere with SSE potentially make heparin a very interesting molecule for SAH management. Some studies report a reduction of DCI in patients treated with intravenous unfractionated heparin for over 48 h. However, it is not clear whether the clinical benefits derive from systemic effects with reduced occurrence of intracranial vessel microthrombosis or from decreased SSE. With respect to the latter hypothesis, it is likely that the effect of heparin is related to the amount of systemic drug delivered in the CSF through the disrupted BBB. To our knowledge, it is not known what concentrations heparin can reach in CSF by intravenous administration in the context of SAH and if the effect of heparin can be dose-dependent. If a relationship between heparin levels in the CSF and improvement of SSE markers or clinical outcomes were confirmed in further clinical studies, new perspectives could open up.

This review aims to underline that SSE is primarily compartmental CSF events as demonstrated by the production and release in the CSF itself of many mediators and biologically active molecules. Definitely, SAH consists of large or massive blood extravasation in the subarachnoid space. At the time of aneurysm rupture the basal cisterns and subarachnoid spaces are normally clean, free from inflammatory events and BBB is intact. It is convincing, as already claimed by other authors, that subsequent events are primarily compartmentalized and pathophysiological pathways occur mainly in the subarachnoid space. It would be legitimate, at this point, to discuss the direct administration of heparin in the CSF: a compartmental therapy for compartmental events.

The CSF compartment is easily accessible as demonstrated by the routine use of ventricular and lumbar subarachnoid catheters. The early intrathecal administration of heparin, after the aneurysm has been closed and secured, could theoretically bring two benefits. The first is associated with e blood clots formation and evolution so that heparin would promote and facilitate the cleaning of cisterns and subarachnoid spaces. The second positive effect might be related to the pleiotropic effects of heparin with the reduction of SSE through the mechanisms listed above. There is no data regarding the administration of heparin in CSF and consequently on its safety. However, clarifying whether the administered systemic heparin is present, and at what concentration, in the CSF of patients with SAH, could allow us to hypothesize a dose to be administered and an adequate dilution. Kole et al. speculate that the route of heparin administration is closely related to the bioavailability of different MW fractions [15]. Subcutaneous absorption decreases with increasing MW [182], whereas greater biological activity is shown by high MW fractions [183]. The direct administration of heparin in the CSF can obviously improve the bioavailability of the drug and allow a greater modulation of the dosage.

The occurrence of bleeding complications, including intracerebral hemorrhage, epidural or subdural hematoma, and external ventricular drain track hemorrhage, was not statistically different in patients with therapeutic and prophylactic heparin doses [15,184]. If the presence of heparin is not associated with a greater risk of bleeding, then the route used to administer should be irrelevant. Furthermore, the intrathecal administration of heparin could avoid the systemic effects and toxicity of the drug itself preserving the patient’s coagulation status. It is more likely that bleeding events, particularly intracranial ones, are linked to alterations in coagulation rather than to events related to the CSF compartment. However, the risks associated with the potential intrathecal administration of heparin remain entirely to be evaluated.

Although the potential role of heparin as an effective treatment for SAH has not been investigated in the context of clinical trials yet, the available literature data and the increasing knowledge on the pathophysiological events related to SAH make the hypothesis intriguing at the least. A research protocol is being designed for the quantification of CSF levels of heparin during systemic administration and for the evaluation of the effect on the occurrence of SSE through the evaluation of markers and mediators compared to a control population. In any case, the evaluation of the role that systemic heparin may have in the patient with SAH can improve the knowledge and scientific background of the pathophysiology of SAH itself. Finally, heparin, as defined by Khattar and James, could really be the Silver Bullet of aneurysmal SAH [181].

## 5. Conclusions

Future improvement in the functional outcome for SAH patients will probably derive from pathophysiological therapy. The cascade of events secondary to SAH is very complex and involves several interconnected and distinct pathways occurring primarily in the CSF compartment and suggesting the need for multiple target therapy both using different drugs combination or multifunctional agents. Heparin may represent a multitarget therapy with an interesting perspective to be investigated in clinical and experimental settings.

## Figures and Tables

**Figure 1 ijms-24-07832-f001:**
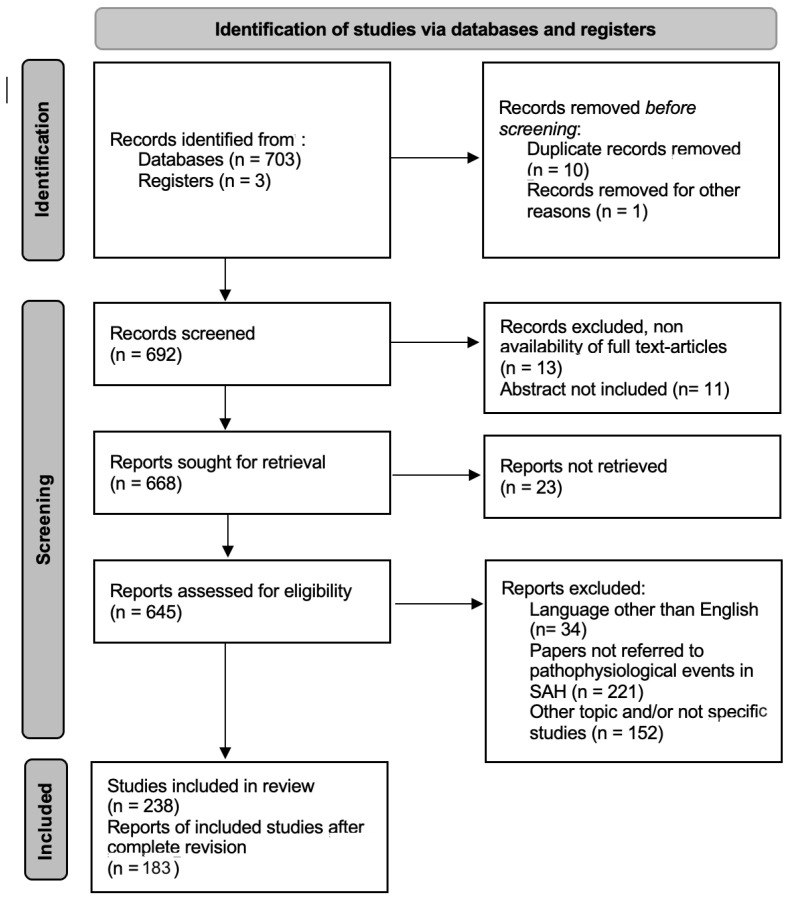
The PRISMA 2020 flow diagram for systematic reviews of “Compartmental cerebrospinal fluid events occurring after subarachnoid hemorrhage” [17].

**Figure 2 ijms-24-07832-f002:**
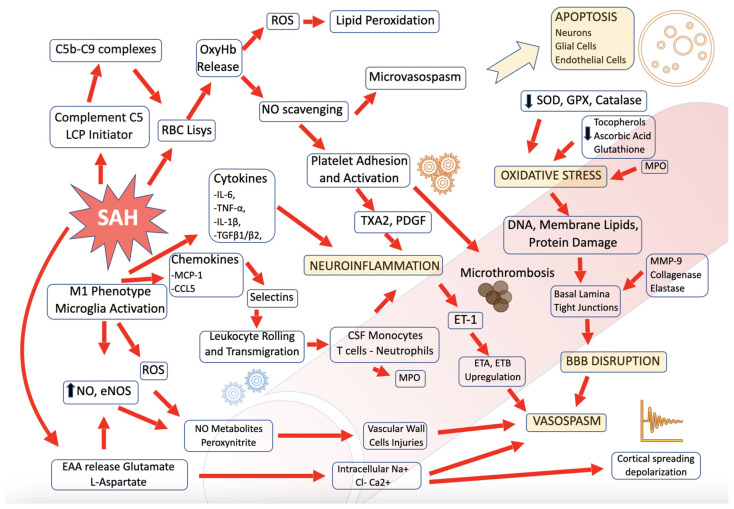
Schematic representation of pathophysiological events occurring in the subarachnoid compartment after SAH.

**Figure 3 ijms-24-07832-f003:**
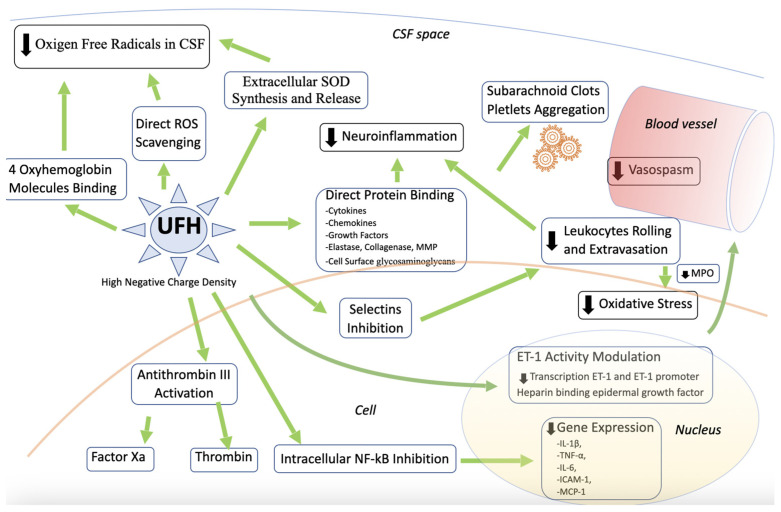
Diagram summarizing the potential effect of heparin on post-SAH pathophysiological events.

## Data Availability

Not applicable.

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
