# Peer review of "Compartmental Cerebrospinal Fluid Events Occurring after Subarachnoid Hemorrhage: An “Heparin Oriented” Systematic Review"

_ijms, 2023, doi:10.3390/ijms24097832_

Round 1

Reviewer 1 Report

1. The introduction would benefit from additional information about subarachnoid hemorrhage (SAH), including a clear definition of what it is, who is most commonly affected, and the risk factors associated with it. It would be useful to provide details about the symptoms, causes, and consequences of SAH, as well as the epidemiology of the condition. Additionally, a discussion of the current treatment options and their limitations could be included to provide context for the study. By incorporating this information, the introduction can better establish the importance of the study and its potential implications for the field. In addition to that the introduction mainly discusses the various secondary events that occur after SAH and their impact on patients' health outcomes. However, it does not provide a comprehensive understanding of the root causes of these events or the molecular mechanisms underlying SAH.

2. One of the inclusion criteria for the clinical studies chosen was that patients were older than 18 years of age. This was chosen to ensure that the studies included adult patients, who are more likely to experience SAH than children. However, it is important to note that SAH can occur at any age. Is there another reason for choosing this age?

3. Although heparin has been used in the treatment of SAH, it is important to consider the potential disadvantages of this treatment. For example, the optimal timing of heparin administration is unclear, as early administration may reduce the risk of vasospasm but increase the risk of bleeding. Therefore, the use of heparin must be carefully monitored and individualized to each patient's specific needs and risks. Adding to that The author mentions bleeding complications associated with heparin use but does not provide a detailed analysis of the potential risks associated with administering heparin directly into the CSF such as neurological complications, including seizures, headaches, and confusion.

4. Heparin can be considered an expensive medication, especially when used over a long period of time. However, the cost can vary depending on factors such as the dosage, frequency of administration, and formulation used. Additionally, the cost of heparin may also vary depending on the healthcare system and country in which it is being used. So you can’t simply say that it is not expensive as written in lines 659, 660.

Author Response

We sincerely thank the Reviewer for suggestions.

1) We modified introduction with more information on SAH and we try to better clarify the aim of the review. This is the first paper suggesting the hypothesis to use intrathecal heparin. The way to clinical practice will be long including experimental mouse model of SAH and discussion on the argument. We suggest in discussion some consideration about a way to better understend role of heparin in SAH treatment. This will be the main implication for the field and prospective on patients outcome impact.

2) Occurrence of SAH in children and pediatric age is very rare. Series on pediatric SAH are few and include syndromic patients and aneurysm considered as congenital. The choice of age over 18 years old is an agreement frequently adopted in studies on SAH addressed to identify more homogeneous population. Furthermore, pathophysiological events in children and pediatric patients may be different related to peculiarity of immune system status.

3) Studies on heparin use in SAH are few. We underline the need for further study at the end of paragraph 3.12. Heparin administration always started, in the analyzed paper, after aneurysm closure and increased bleeding risk is not reported. We agree that risks related to direct CSF administration remain completely to be evaluated. We add this suggestion to discussion

4) We agree with the comment of Reviewer and we remove the consideration about heparin cost

Reviewer 2 Report

This comprehensive review has a number of shortcomings:

1 The term 'CSF' is used without any indication of where it was obtained from. This needs to be corrected.

2. Excitotoxicity needs to be mentioned, especially as there is data in the literature showing elevated concentrations of excitatory amino acids in ventricular CSF following both SAH and traumatic brain injury.

3. Oxidative stress is under represented. This needs to be corrected.

4. Does heparin have antioxidant activity.

5 The conclusion needs to emphasise the need to target multiple pathophysiological process through polypharmacy or monotherapy with multifunctional compounds.

6. A diagram that summarises the multiple pathophysiological processes would be helpful.

Author Response

Many thank for the helpful suggestions

1)We specified the term CSF to be expression of cerebrospinal fluid

2)Thank you for the valuable suggestion. Actually by extending our research we have pointed out that excitotoxicity is an important mechanism in secondary neuronal injury after traumatic brain injury and SAH and the increased level of excitatory amino acids representing a potential signaling of brain damage and predictor of outcome.

3)We agree that oxidative stress need to be better underlined. So we provide a more representation in the text.

4)As reported in paragraph 3.12 heparin present antioxidant activity through direct scavenging of free radicals and induction of SOD production in CSF. Indirectly heparin reduce free radical production as consequence of oxyhemoglobin binding and inactivation. Antioxidant activity of heparin was enhanced in the text and trough a diagram on heparin actions

5)We provide, as suggested, to specify in conclusion need for multiple target therapy

6)A diagram with pathophysiological pathways as been added and a diagram on heparin potential actions is also included

Round 2

Reviewer 1 Report

All is good, thank you for considering all comments in order to improve your work. 

Author Response

Many for the kind suggestions that have certainly made it possible to improve the paper.

Reviewer 2 Report

The authors have addressed my comments, but not adequately.

Specifically:

1. they should mention that both aspartate and glutamate are elevated in ventricular CSF following traumatic brain injury, where hemorhage also occurs -  https://pubmed.ncbi.nlm.nih.gov/7703404/

2. Section 3.7 fails to mention L-aspartate and the NMDA receptor - this needs to be corrected.

3. Excitotoxiciity needs to be incorporated into the new figures

Author Response

We thank again Reviewer for further suggestions

1. Suggested reference has been inserted in text.

2. Section 3.7 has been extended with reference to L-Aspartate and NMDAR antagonist 

3. Excitotoxicity has been added to figure 2 (pathophysiological events) but not in figure 3. No evidence are available about UFH in front of EAAs and glutamate in particular.